# Exploring the dimensionality of the 5C positive youth development very short form using Rasch measurement theory in Swedish upper secondary school contexts

Johanna Bergman[1]*, Maria Haak[1], Petra Nilsson Lindström[1], Åsa Bringsén[1], Albert Westergren[1,2]

**1** Faculty of Health Sciences, Kristianstad University, Kristianstad, Sweden, **2** Department for Research, Development and Education, Management for Psychiatry and Habilitation, Region Skåne, Lund, Sweden

* johanna.bergman@hkr.se

## Abstract

The 5C Positive Youth Development Very Short Form (PYD-VSF) was designed as a user-friendly tool to assess positive youth development. This study examined the dimensionality using Rasch measurement theory on data from 430 upper secondary school pupils in southern Sweden. A Swedish version of the tool was developed through dual-panel translation and cognitive interviews. Findings from psychometric testing indicated multidimensionality, showing support for a post hoc two-dimensional structure with the following components: Self-worth (*Competence*, *Confidence,* and *Connection*) and Pro-social (*Character* and *Caring*). These two constructs were distinct yet related, suggesting that Self-worth may be a prerequisite for Pro-social development. Systematic gender variance was also observed, although with a negligible impact on score interpretation. The identified constructs align with ongoing discussion about agency and resilience in relation to young persons' health. Further research is recommended within the PYD framework, with priority given to the use of standardised and validated instruments to clarify how these constructs may both strengthen young persons in the present and buffer against future adverse outcomes.

## Introduction

The youth era of life is commonly described as intense, dynamic, and challenging. Often, society views this era as problematic and potentially conducive to increased risk behaviours [1]. However, this risk perspective is dominated by an authority perspective where youths could be considered as "problems to be solved" [2]. To challenge this perspective, a growing field of studies within positive youth development (PYD) has emerged [3–5]. The PYD theory explains youth development as a mutually influential relation between individuals and different aspects of their contexts,

**Data availability statement:** Data cannot be shared publicly because of ethical decision from The Swedish Ethical Review Authority. The data set has a DOI: https://doi.org/10.5878/w9wy-ak07 but restrictions apply. Data are available from author if permission is obtained from The Swedish Ethical Review Authority (contact registrator@etikprovning.se) for researchers who meet the criteria for access to confidential data.

**Funding:** Funded by Kristianstad University. The funder had no role in study design, data collection and analysis, decision to publish, or preparation of the manuscript.

**Competing interests:** The authors have declares that no competing interests exist.

often referred to as a relational developmental system (RDS) [6–8]. This positive and dynamic perspective has captured researchers' and scholars' interest as they seek to further understand what underpins PYD, in order to implement successful youth programmes in various domains that promote it. However, literature reviews have indicated that varied definitions and methodologies have been applied in research, making it difficult to determine outcomes and effects [5,9–12]. Consequently, the field of PYD risks becoming fragmented, which may hinder its theoretical clarity and practical application.

In the vast body of literature on PYD, one of the earliest and most commonly used models is the 5C model, where five dimensions of PYD have been identified that together reflect positive youth development: Competence, Confidence, Character, Caring, and Connection [13]. This model has informed the development of the PYD 5C instrument [13], which has been used internationally to measure youth development [3,7,14,15]. The original PYD 5C instrument consists of 78 items [13], with a 38-item short form (PYD-SF) and a 17-item very short form (PYD-VSF) created to enhance practical usability [16]. Accessible tools such as these are essential for implementation in various settings aiming to improve PYD, for example, in schools or sports. Psychometric analysis of these instruments in various contexts has produced differing results, with studies identifying between two and seven dimensions using exploratory and confirmatory factor analysis [15,17,18]. This inconsistency complicates theoretical interpretations of the 5C model and may challenge its utility in practice [19,20].

Assessing the dimensionality of the instruments is an important step to improve practical utility. Unidimensionality means that the scale items should represent a common latent variable, hence undimensionality refers to clarification of what is being measured. If the instrument is not measuring the variable it is supposed to, it could be argued that the total score is technically invalid [21]. Previous research has confirmed that a higher-order construct is not recommended using PYD 5C instruments, as the five components do not appear to reflect a single underlying dimension. Instead, a bifactor model has been proposed [22,23]. Exploratory structural equation modelling has also been utilised and is considered an even more suitable method for testing the factor structure of the 5C [24]. These studies are based on psychometric analysis commonly referred to as Classical Test Theory (CTT) [25]. CTT is dependent on strong data, presumptions, and sample size and is based on common variance and/or differences between persons at group level rather than at person level [26–31]. However, the key aspect is that CTT mainly relies on correlations, which is both a strength and limitation. It allows for straightforward statistical analysis and has been widely used in test development, but it may also overlook the interpretability and meaningfulness of the latent variable [21,25,32]. Furthermore, analysis is usually based on ordinal data, which can be problematic since techniques like Pearson correlations assume interval-level data. This mismatch may lead to inaccurate estimates of relationships between items, and both exploratory and confirmatory factor analysis are traditionally based on Pearson correlations [33]. A more modern and robust psychometric approach to exploring the dimensionality of the 5C instruments is the Rasch measurement theory (RMT).

RMT is based on probability analysis rather than correlations, and items can be explored in detail in terms of dimensionality, local dependencies, response category function, hierarchical orders, and multicollinearity [25,34,35]. The Rasch model within this psychometric approach determines model fit by separately placing individuals and items on a common logit scale (at the interval level) that ranges from minus to plus infinity, with the average item location set to zero. The success of the measurement is evaluated by analysing how well the observed data align with the Rasch model's expectations. If the data sufficiently conform to the model, linear measurement and invariant comparisons become possible; hence, precise measurements at personal level also become possible [34].

The aim of this study was to adapt PYD-VSF to a Swedish context and explore the dimensionality and psychometric properties using RMT based on a survey of Swedish upper secondary school students (age 15–18).

## Method

### The PYD-VSF instrument

To meet the demands of an accessible instrument for practical settings, and to ensure it is user-friendliness the study used the very short form of PYD (PYD-VSF) with 17 items [16]. The 5Cs, which are central components of the PYD framework, are defined as follows: *Competence* refers to the positive view of one's actions in specific areas, including social, academic, cognitive, health, and vocational. *Confidence* is defined as one's internal sense of overall positive self-worth. *Connection* relates to the positive bonds between people and institutions and reflects the mutually influential interactions that form the relationship between the individual and his or her peers, family, school, and community. *Character* is conceptualised as respect for societal and cultural norms, possession of standards for moral behaviours, a sense of right and wrong, and integrity. *Caring* refers to being cared for and caring for others and has been related to one's sense of sympathy and empathy for others [13,36]. Each C is a subscale consisting of three or four items that are argued to function on their own using an average of included items or adding up all 17 items and calculating a mean representing the overall PYD [16]. All items are scored on a Likert scale ranging from one to five, as follows: "Strongly disagree [1] – Disagree [2] – Neither disagree nor agree [3] – Agree [4] – Strongly agree [5]". Higher scores on all items reflect a greater degree of the respective C or overall PYD.

The procedure was divided into two separate phases. First, the translation and cultural adaptation of PYD-VSF. Second, the psychometric testing of the Swedish version of PYD-VSF.

### Phase one: Translation and cultural adaptation of the PYD-VSF

The translation and cultural adaptation process was inspired by Hunt [37] and Westergren [38] using a dual-panel methodology. An initial bilingual panel consisting of six professional representatives (representing the fields of health, pedagogy, social science, and English) discussed the translation from English to Swedish and backwards. As the questionnaire is intended to have non-technical language, the most agreed-upon translation within the panel was reviewed by six lay people (three women, three men aged 40−45) outside of academia to ensure the clarity and appropriateness of the language [37]. The questionnaire was then added to the Evasys digital survey programme and used to conduct cognitive interviews with the target group (persons aged 15−18 years). Fifteen pupils (seven girls, eight boys) from two different upper secondary schools (one rural and one urban) were asked to participate in cognitive interviews [39]. The pupils were provided with an information letter that included the diary number of the ethical approval and stated that participation was optional, and they could choose to withdraw without explanation at any time. After reading the information letter, informed consent was obtained verbally. This verbal consent was further confirmed by the participants' active engagement in the interview. First, the survey was administered under time surveillance to evaluate the average completion time (no answers were submitted). After completion, they were asked questions regarding phrasing, user-friendliness, and their understanding of concepts, and were asked for other feedback they considered relevant. No interviews were recorded; instead, notes were taken and used for further discussion, in which the developers of the original PYD-VSF were also

consulted. The procedure of the cognitive interviews was approved by the Swedish Ethical Review Authority (Diary number 2024-01088-01).

### Phase two: Psychometric testing of the PYD-VSF

**Survey procedure.** A total of 42 upper secondary schools with pupils aged 15–18 in the southern region of Sweden were invited to participate from 17th April 2024–1st October 2024 in a cross-sectional study using the PYD-VSF. Both rural and urban locations within the region were included. Schools were provided with a digital link to Evasys and a QR code to distribute the questionnaire to their pupils either via internal school platforms or posters. Participation was optional, and upon using the link or QR code, a letter of information about the study appeared along with the diary number of the ethical approval. The information letter stated the estimated completion time (5–10 minutes), and provided contact details to receive appropriate support if questions arose as a consequence of participation. The letter also stated their right to withdraw from the study after opening the link, and that actively clicking "submit" after completion of the questionnaire signified informed consent. The procedure of informed consent was approved by the Swedish Ethical Review Authority (Diary number 2024-01088-01), and according to the Ethical Review Act section 13–19 [40], guardians' consent is not required for participants aged 15–18. No personal information was collected; hence, pupil anonymity was maintained during and after the data collection.

**Data analysis.** The Rasch measurement analysis was conducted using RUMM2030 Professional Edition [41]. The analysis was performed in three steps. First, the initial 17-item scale of PYD-VSF was explored. Second, the original five-dimensional model of PYD-VSF using the subscales of *Competence, Confidence, Character, Caring, and Connection* was explored. The subscales were also analysed in a subsequent subtest analysis. A subtest analysis combines items related to one subscale into one single item in the analysis [42,43]. This enables preservation of the total domain score while absorbing potential local dependency in the analysis [42]. The third step was a post hoc analysis of a two-dimensional model in response to the findings from the initial analysis of the original PYD-VSF 17-item scale and the five-dimensional subtest analysis. In the analyses, compliance with the Rasch model was evaluated with respect to: 1) targeting, 2) item and response category monotonicity, 3) item-trait interaction, 4) item-fit and local item independence (LID), 5) reliability, 6) person fit, 7) differential item functioning (DIF), and 8) unidimensionality. These eight aspects (as defined in Table 1) were applied as appropriate, depending on whether the analyses concerned all items, subscales, or subtests.

Any problems found during the different analyses were approached with a discussion of how these could be resolved, such as merging response categories or splitting items with DIF. Eventual DIF person locations (logit measures) on the PYD-VSF were adjusted for DIF through item splitting by the person factor and compared with those derived from the original (non-adjusted) items, which then were anchored using DIF-free item calibrations from the DIF-adjusted scale [44].

Finally, provided that the data are sufficiently consistent with" the perfect" Rasch model, the qualitative (ordinal) raw scores can be transformed into interval-level measures. When a satisfactory model fit was achieved, the resulting Rasch person estimates, expressed in logits, were rescaled to align with the range of the original raw scores. Such interval-level measures enable quality-assured comparisons at the person level, both between persons and over time [48].

## Results

### Phase 1: Translation and cultural adaptation of PYD-VSF

Phase 1 is illustrated in Fig 1. After the cognitive interviews, minor changes in wording were made to item(i)15 and 16. Items 6, i15, and i16 were discussed with the developer of PYD-VSF to confirm that the translation still captured the same phenomenon.

**Table 1. Overview of Rasch measurement analyses, definitions and interpretations.**

| Analysis | Definition | Interpretations |
|---|---|---|
| **Targeting** | The match between item difficulty and person ability distributions [42]. | Mean (logit) and Standard deviation (SD) of Person–Item distribution. It should be as close to 0 as possible to indicate good measure precision and good prerequisites for scale evaluation. |
| **Item and response category monotonicity** | The probability of endorsing higher item responses increases monotonically with the latent trait [34,42]. | Investigating threshold maps and category probability curves. |
| **Item-trait interaction** | $\chi^2$-based fit statistic that tests whether the hierarchical ordering of items is consistent across different levels of the latent trait [42]. | A significant result after Bonferroni adjustment ($p < 0.05$) indicates misfit, i.e., that items function differently depending on a person's ability. |
| **Item-fit and local item dependency (LID)** | How well an item matches Rasch model expectations and whether an answer to an item is dependent upon the answer of another item [34,45]. | Using fit residuals at item level (acceptable range: −2.5 to +2.5) and Bonferroni-adjusted chi-square statistics ($p < 0.05$), along with item characteristic curve inspection.<br>LID: Item residual correlations using Yen's Q3*. Item pair correlations where Yen's Q3* $> 0.2$ was considered indicative of potential dependency. |
| **Reliability** | Person Separation Index and Cronbach alpha for subscales (PSI/ α). PSI is a Rasch-based reliability coefficient indicating how well the scale can distinguish between individuals at different levels of the latent trait [46]. | PSI/α $> 0.7$ |
| **Person fit** | How well a person's response matches Rasch model expectations [42]. | Using fit residuals at the person level (acceptable range: −2.5 to +2.5). Reported at group level % of person fit residuals within +/- 2.5. |
| **Differential item functioning (DIF)** | Same trait level, different item probability across groups [47]. | Subgroups identified: school year and gender. Number of items with DIF, and DIF characteristics. |
| **Unidimensionality** | All items measure a single underlying latent trait [21,42,43]. | Principal component analysis (PCA) of item residuals followed by paired t-tests of person estimates, with ≤5% significant tests (Agresti-Coull 95% CI) indicating unidimensionality. Item groups for t-tests were defined by residual loadings $\geq +0.3$ or $\leq -0.3$ on the first component, reflecting potential subdimensions. If ≤5% of tests are significant, unidimensionality is supported. At least 12 items are recommended.<br>Subtest: Subtest analyses estimated indices A, C, and r, reflecting shared variance, unique variance, and latent correlation, respectively. Unidimensional scales yield high A and r, and low C. |

## Phase 2: Psychometric testing of the PYD-VSF

A total of 430 upper secondary school pupils participated in the study (59.5% girls, 38.8% boys, 0.7% non-binary, 0.7% other, and 0.2% missing). The response pattern per item and missing data are presented in Table 2. In general, there was a low percentage of missing answers on separate items (0–1.4%), and "strongly disagree" and "disagree" were not used as frequently as the other response categories. In total, 395 pupils (92%) submitted complete forms, and these were included in the RMA.

An overview of key RMA analyses can be found in Table 3. Table 3 serves as a point of reference throughout the Results section, where each analysis is described in more detail.

The PYD-VSF 17-item scale is first presented, followed by the five subscales, the five-dimensional the two post hoc subscales, and two-dimensional subtest. All person-item distribution graphs of the five subscales can be found in S1 Appendix.

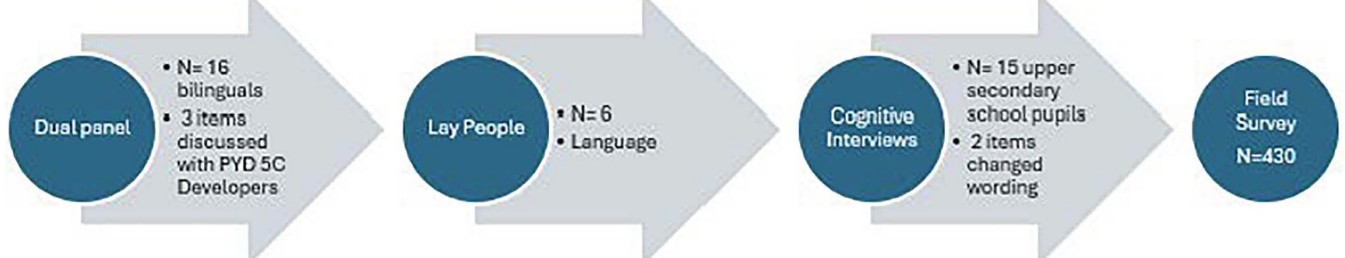

**Fig 1. Overview of the translation and cultural adaptation process of PYD-VSF.**

### Step one: PYD-VSF original scales

**17-item scale.** The person–item distribution indicated a mismatch between item difficulty and respondents' ability (Fig 2). The scale failed to represent higher levels of experienced PYD where the sample was located (logits: mean 0.936, SD 1.012). No a priori expectation of a certain theoretical hierarchical order was expected, and the item order (Table 4) did not follow the subscales in general, except for Caring (i11, i22, and i13).

The analysis showed that i1, i2, i8, i9, i10, i11, i12, i13, i14, i15, i16, and i17 had disordered thresholds where the response "disagree" did not function as expected.

The summary item-trait interaction was significant ($\chi2 = 319$, $p > 0.00$, indicating misfit to the Rasch model.

In terms of item-fit, i3 and i5 had residuals >2.5 indicating multidimensionality, and i7, i15, and i16 had residuals < −2.5 indicating local dependencies (Table 4). Furthermore, Yen's Q3* indicated that nine item pairs had residual correlations (>0.2) where the correlations followed the pattern of being locally dependent on another item within a specific subscale of the 5C subscales.

The PSI for the scale was 0.874. The PSI can be affected by the multiple signs of local dependencies (fit residual < −2.5) at item level (i7 i15 and i16), and high residual correlation between item pairs, which can cause an artificially inflated reliability, without reflecting true increases in measurement precision [42].

In terms of person fit, 10% of the sample's responses did not match the expected Rasch model, and six items (i3, i4, i6, i7, i11, i12, and i13) indicated a uniform DIF in terms of gender. For i3, i4, i6, and i7, uniform DIF was identified where boys responded consistently higher than girls. For i11, i12, and i13, the opposite was found, with girls responding consistently higher than boys.

The PCA and t-test protocol showed that 14.43% of the t-tests were significant (Agresti-Coull 95% CL 11.29%−18.26%), indicating that the scale is multidimensional.

As 12 items showed disordered thresholds, an analysis was conducted, rescoring the dysfunctional "strongly disagree" and "disagree" response categories to one response category in all items. This did not resolve the 12 disordered thresholds; the item-trait interaction ($\chi = 205.86$) was still significant (PSI 0.855) but the person misfit decreased to 7.3%.

### Competence subscale analysis

The person–item distribution indicated a mismatch between item difficulty and respondents' ability (S1 Appendix) where the scale failed to represent higher levels of experienced *Competence* (logits: mean 0.874, SD1.042). This indicated that persons in the sample experienced higher levels of *Competence* than the items captured.

**Table 2. Theoretically grouped items, item response pattern, and missing items (n = 430).**

| Item(i) number and label | Response pattern, n (%) | | | | | |
|---|---|---|---|---|---|---|
| | Strongly disagree | Disagree | Neither agree nor disagree | Agree | Strongly agree | Missing |
| *Competence* | | | | | | |
| i1. I have a lot of friends | 16 (3.7%) | 22 (5.1%) | 74 (17.2%) | 13 (30.5%) | 187 (43.5%) | 0 (0%) |
| i2. I do very well in my class work at school | 13 (3%) | 25 (5.8%) | 117 (27.2%) | 179 (41.6%) | 94 (21.9%) | 2 (0.5%) |
| i3. I am better than others my age at sport | 49 (11.4%) | 7 (18.1%) | 109 (25.3%) | 117 (27.1%) | 75 (17.4%) | 2 (0.5%) |
| *Confidence* | | | | | | |
| i4. I am happy with myself most of the time | 21 (4.9%) | 73 (17%) | 92 (21.4%) | 147 (34.2%) | 94 (21.9%) | 3 (0.7%) |
| i6. I really like the way I look | 39 (9.1%) | 7 (18.4%) | 104 (24.2%) | 126 (29.3%) | 80 (18.6%) | 2 (0.5%) |
| i7. All in all, I am glad I am me | 15 (3.5%) | 33 (7.7%) | 90 (20.9%) | 156 (36.3%) | 132 (30.7%) | 4 (0.9%) |
| *Character* | | | | | | |
| i5. I hardly ever do things I know I shouldn't do | 31 (7.2%) | 73 (17%) | 101 (23.5%) | 136 (31.6%) | 88 (20.5%) | 1 (0.2%) |
| i8. I want to make the world a better place to live | 13 (3%) | 17 (4%) | 78 (18.1%) | 159 (37%) | 160 (37.2%) | 3 (0.7%) |
| i9. I accept responsibility for my actions when I make a mistake or get in trouble | 6 (1.4%) | 8 (1.9%) | 48 (11.2%) | 198 (46%) | 168 (39.1%) | 2 (0.5%) |
| i10. I enjoy being with people from different cultures and backgrounds | 21 (4.9%) | 24 (5.6%) | 76 (17.7%) | 154 (35.8%) | 151 (35.1%) | 4 (0.9%) |
| *Caring* | | | | | | |
| i11. When I see someone being taken advantage of, I want to help them | 8 (1.9%) | 6 (1.4%) | 71 (16.5%) | 157 (36.5%) | 184 (42.8%) | 4 (0.9%) |
| i12. When I see someone being picked on, I feel sorry for them | 11 (2.6%) | 7 (1.6%) | 50 (11.6%) | 128 (29.8%) | 231 (53.7%) | 3 (0.7%) |
| i13. When I see another person who is hurt or upset, I feel sorry for them | 11 (2.6%) | 8 (1.9%) | 49 (11.4%) | 136 (31.6%) | 224 (52.1%) | 2 (0.5%) |
| *Connection* | | | | | | |
| i14. I receive a lot of encouragement at my school | 14 (3.3%) | 29 (6.7%) | 149 (34.7%) | 167 (38.8%) | 65 (15.1%) | 6 (1.4%) |
| i15. I am an important person for my family or equivalent | 8 (1.9%) | 7 (1.6%) | 61 (14.2%) | 131 (30.5%) | 219 (50.9%) | 4 (0.9%) |
| i16. I feel like an important member of my social spheres | 10 (2.3%) | 22 (5.1%) | 92 (21.4%) | 152 (35.3%) | 150 (34.9%) | 4 (0.9%) |
| i17. I feel my friends are good friends | 13 (3%) | 17 (4%) | 56 (13%) | 134 (31.2%) | 206 (47.9%) | 4 (0.9%) |

The analysis also revealed that i1 and i2 had disordered thresholds, where the response category "disagree" did not function as expected for i1, and both the response categories "strongly disagree" and "disagree" did not function as expected for i2.

Table 5 summarises all item-fit statistics for the five subscales, and for the subscale of *Competence,* no item indicated multidimensionality or local dependency, and Yens' Q3* indicated no high residual correlations between item pairs.

The summary item-trait interaction was significant ($\chi2 = 29.59$, $p < 0.01$), suggesting an overall misfit to the Rasch model. The PSI for the scale was 0.455.

In terms of person fit, 9% of the sample's responses did not match the expected Rasch model, and i2 indicated a uniform DIF in terms of gender. Girls consistently scored higher than boys.

**Table 3. Overview of all RMA analyses of the 17 items, the five- and two-dimensional scales.**

| Analysis | Inter-pretation | 17-item scale | Sub. Comp-etence | Sub. Confi-dence | Sub. Charac-ter | Sub. Caring | Sub. Connec-tion | Subtest-5Dim | Sub. Self-worth | Sub. Pro-social | Subtest-2Dim |
|---|---|---|---|---|---|---|---|---|---|---|---|
| **Targeting** Mean log | More distant to 0=worse | 0.936 | 0.874 | 1.189 | 0.825 | 3.270 | 1.497 | 0.590 | 0.944 | 1.234 | 0.396 |
| (SD) | | | (1.012) | (1.042) | (2.427) | (0.910) | (2.297) | (1.580) | (0.683) | (1.217) | (1.104) | (0.428) |
| **No. of items violating monotonicity** | Disordered thresholds | 12 | 2 | 0 | 3 | 1 | 3 | – | 6 | 6 | – |
| **Item-trait interaction** | $\chi^2$ p<0.05 | 319 | 29.59 | 30.02 | 43.57 | 45.29 | 57.29 | 55.61 | 166.22 | 184.81 | 14.60 |
| | | p<0.00 | p<0.01 | p<0.01 | p<0.00 | p<0.00 | p<0.00 | p<0.00 | p<0.00 | p<0.00 | p=0.15 |
| **No. of items with misfit** | FR<−2.5 p<0.05[a] | 5 | 0 | 0 | 0 | 0 | 0 | 0 | 3 | 4 | 0 |
| **No. of items with misfit** | FR>+2.5 p<0.05[a] | 0 | 0 | 0 | 0 | 1 | 0 | 0 | 0 | 0 | 0 |
| **No. of item pairs with local dependency** | Yen's Q3*>0.2 | 9 | 0 | 0 | 0 | 0 | 0 | 0 | 4 | 1 | 0 |
| **Reliability (PSI/α)** | <0.70 | 0.874 | 0.455 | 0.832 | 0.481 | 0.671 | 0.697 | 0.884 [b] | 0.850 | 0.715 | 0.884 [b] |
| **Person fit** | MD if ≥5% misfit[a] | 10% | 9% | 15% | 9% | 37% | 11% | 6% | 9% | 13% | 3% |
| **DIF school year** [a] | No. of items with DIF | 0 | 0 | 0 | 0 | 0 | 0 | 0 | 0 | 0 | 0 |
| **DIF gender** [a] | No. of items with DIF | 6 | 1 | 0 | 0 | 0 | 0 | – | 4 | 3 | – |
| **Unidimensionality PCA/t-test, %** | ≥5% sign | 14.43% | – | – | – | – | – | 6.08% | – | – | 1.52% |
| (95% CI interval) | | 11.3-18.3 | | | | | | 4.1-8.9 | | | 0.6-3.4 |
| **Unidimensionality** (subtest analysis only) | UD if: | – | – | – | – | – | – | | – | – | |
| | C (low) | | | | | | | 0.710 | | | 0.656 |
| | r (high) | | | | | | | 0.665 | | | 0.699 |
| | A (high) | | | | | | | 0.923 | | | 0.801 |

FR = Fit Residual. UD=Unidimensionality [a] Significant after Bonferroni adjustment [b] Cronbach´s alpha.

## Confidence subscale analysis

A lack of fit was reflected in the person–item distribution, indicating a mismatch between item difficulty and respondents' ability (S1 Appendix), where the scale failed to represent higher levels of *Confidence* (logits: mean 1.189, SD 2.427). This indicates that persons in the sample experienced higher levels of *Confidence* than the items captured.

The analysis revealed no disordered thresholds, and the item-fit (Table 5) did not show any signs of multidimensionality or local dependencies. Yen's Q3* indicated no high residual correlations between item pairs either.

The summary item-trait interaction was significant ($\chi2=30.02$, p<0.01), indicating an overall misfit to the Rasch model. The PSI for the scale was 0.832.

In terms of person fit, 15% of the sample's responses did not match the expected Rasch model, and no DIF in terms of school year or gender was found.

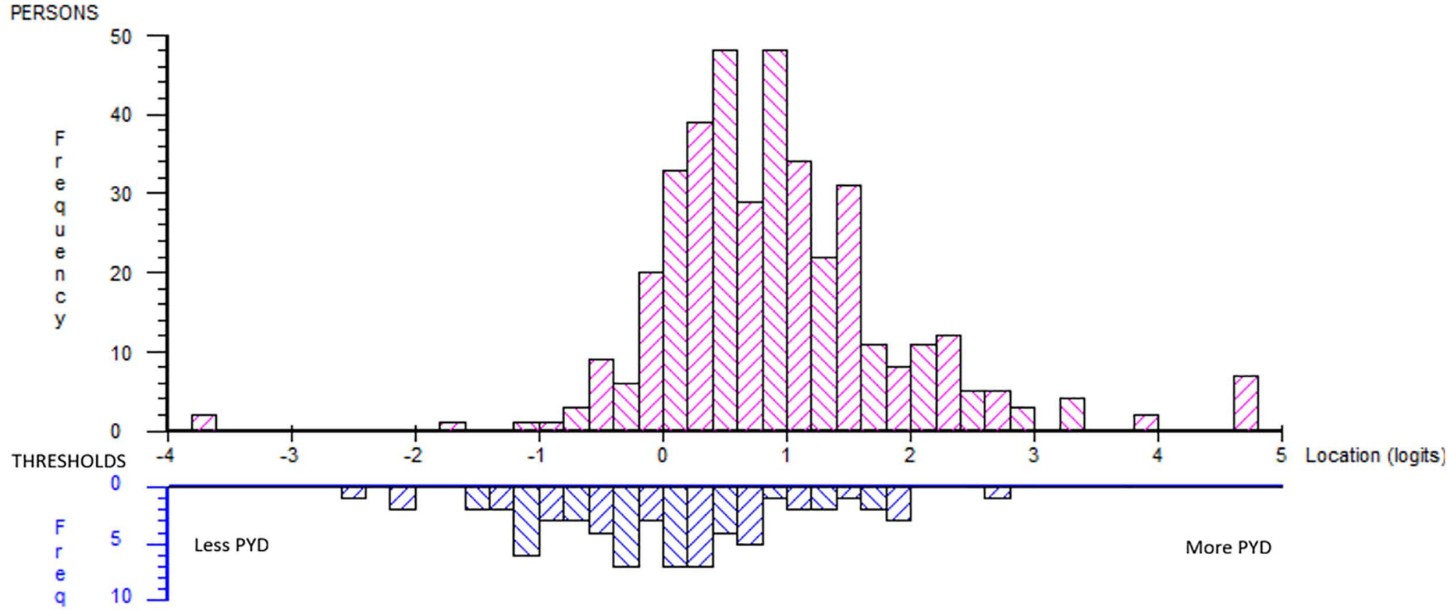

**Fig 2. Person-item distribution original 17-item scale.**

**Table 4. Item-fit statistics original 17-item scale in location order.**

| Item (i) | Location | SE | FitResid[a] | $\chi^2$ | Prob |
|---|---|---|---|---|---|
| i9 | −0.571 | 0.075 | −0.165 | 5.089 | 0.405075 |
| i15 | −0.515 | 0.068 | −3.584 | 33.585 | 0.000003 |
| i11 | −0.425 | 0.069 | −0.443 | 12.018 | 0.034547[b] |
| i12 | −0.356 | 0.064 | −0.624 | 6.813 | 0.234922 |
| i13 | −0.341 | 0.063 | −0.079 | 6.514 | 0.259370 |
| i17 | −0.266 | 0.060 | −0.411 | 2.310 | 0.804794 |
| i16 | −0.186 | 0.063 | −2.945 | 31.317 | 0.000008 |
| i1 | −0.092 | 0.057 | −0.502 | 4.778 | 0.443587 |
| i8 | −0.078 | 0.060 | 0.763 | 8.376 | 0.136689 |
| i7 | −0.006 | 0.060 | −2.819 | 23.978 | 0.000220 |
| i10 | 0.099 | 0.055 | 1.379 | 3.436 | 0.633107 |
| i2 | 0.149 | 0.062 | 1.36 | 0.922 | 0.968623 |
| i4 | 0.358 | 0.055 | 0.139 | 8.508 | 0.130355 |
| i14 | 0.361 | 0.066 | −1.141 | 18.91 | 0.001999 |
| i5 | 0.514 | 0.051 | 8.837 | 101.645 | 0.000000 |
| i6 | 0.647 | 0.052 | −0.019 | 4.325 | 0.503655 |
| i3 | 0.710 | 0.050 | 6.039 | 46.789 | 0.000000 |

[a]Fit Residual (FitRes) above 2.5 indicating multidimensionality. FitRes below −2.5 indicating local dependency

[b]Not significant after Bonferroni adjustment.

**Table 5. Overview of item-fit statistics for separate subscale analyses. Items presented in location order.**

| Subscale | Item (i) | Location | SE | FitResid[a] | χ2 | Prob |
|---|---|---|---|---|---|---|
| Competence | i1 | −0.349 | 0.056 | −0.768 | 13.967 | 0.015824 |
| | i2 | −0.113 | 0.062 | 1.057 | 8.526 | 0.129538 |
| | i3 | 0.462 | 0.051 | 0.614 | 7.095 | 0.213641 |
| Confidence | | | | | | |
| | i7 | −0.755 | 0.087 | −0.597 | 15.201 | 0.009539 |
| | i4 | 0.050 | 0.084 | 0.883 | 7.641 | 0.177143 |
| | i6 | 0.705 | 0.081 | 0.411 | 7.179 | 0.20764 |
| Character | | | | | | |
| | i9 | −0.501 | 0.071 | 0.003 | 7.46 | 0.188606 |
| | i8 | −0.055 | 0.057 | −1.367 | 15.672 | 0.007848 |
| | i10 | 0.080 | 0.053 | −0.869 | 10.957 | 0.052232 |
| | i5 | 0.476 | 0.050 | 2.059 | 9.479 | 0.091417 |
| Caring | | | | | | |
| | i11 | −0,9120 | 0.112 | 3.340 | 21.433 | 0.000087 |
| | i12 | 0.446 | 0.104 | −2.075 | 16.378 | 0.000949 |
| | i13 | 0.466 | 0.103 | −2.323 | 7.487 | 0.057892 |
| Connection | | | | | | |
| | i15 | −0.498 | 0.076 | −1.487 | 16.105 | 0.002883 |
| | i17 | −0.178 | 0.069 | 0.010 | 9.427 | 0.051276 |
| | i16 | −0.044 | 0.074 | −2.266 | 16.496 | 0.002422 |
| | i14 | 0.729 | 0.075 | 1.902 | 15.262 | 0.004189 |

[a]Fit Residual (FitRes) above 2.5 indicating multidimensionality. FitRes below −2.5 indicating local dependency.

## Character subscale analysis

A lack of fit was reflected in the person–item distribution, indicating a mismatch between item difficulty and the respondents' ability (S1 Appendix) where the scale failed to represent higher levels of *Character* (logits: mean 0.825, SD 0.910). This indicates that persons in the sample experienced higher levels of *Character* than the items captured.

The analysis showed disordered thresholds for i8, i9 and i10. The response "disagree" did not function as expected for i8 and i10. For i9, neither the response "strongly disagree" nor "disagree" functioned as expected.

Table 5 shows that no items showed item misfit in terms of multidimensionality or local dependency, and Yen's Q3∗ indicated no high residual correlations between item pairs.

The summary item-trait interaction was significant (χ2 = 43.57, p < 0.00), indicating an overall misfit to the Rasch model. The PSI for the scale was 0.481.

In terms of person fit, 9% of the sample's responses did not match the expected Rasch model and no DIF in terms of school year or gender was found.

## Caring subscale analysis

A lack of fit was reflected in the person–item distribution, indicating a mismatch between item difficulty and the respondents' ability (S1 Appendix), where the scale failed to represent higher levels of *Caring* in the sample (logits: mean 3.270, SD 2.297).

The analysis showed disordered thresholds in i12. The response "disagree" did not function as expected for this item.

The summary item-trait interaction was significant (χ2 = 45.29, p < 0.00), indicating an overall misfit to the Rasch model.

Table 5, presenting the item-fit statistics, indicates multidimensionality for i11 but no local dependencies. Yen's Q3* indicated no high residual correlations between item pairs.

The PSI for the scale was 0.671.

In terms of person fit, 37% of the sample's responses did not match the expected Rasch model, and no DIF in terms of school year or gender was found.

### Connection subscale analysis

A lack of fit was reflected in the person–item distribution, indicating a mismatch between item difficulty and the respondents' ability (S1 Appendix), where the scale failed to represent higher levels of experienced *Connection* in the sample (logits: mean 1.497, SD 1.580).

The analysis showed disordered thresholds in i14, i15 and i17. The response "disagree" did not function as expected for these items.

The summary item-trait interaction was significant ($\chi2 = 57.29$, $p < 0.00$), indicating an overall misfit to the Rasch model.

Table 5 presents the item-fit statistics, where no items showed signs of multidimensionality or local dependency, and Yen's Q3* indicated no high residual correlations between item pairs.

The PSI for the scale was 0.697.

In terms of person fit, 11% of the sample's responses did not match the expected Rasch model, and no DIF in terms of school year or gender was found.

### Step two: Five-dimensional subtest analysis

This analysis was performed as the individual scales failed to provide model fit and had poor targeting and precision. We combined items within the 5C, i.e., five subscales, into subtests (superitems/testlets) and treated each subtest as a single item in the analysis.

The person–item distribution (logits; mean 0.590, SD 0.683) indicated a rather good precision between item difficulty and the respondents' ability (Fig 3). The graph illustrates a minor gap in terms of items representing the higher end of the continuum.

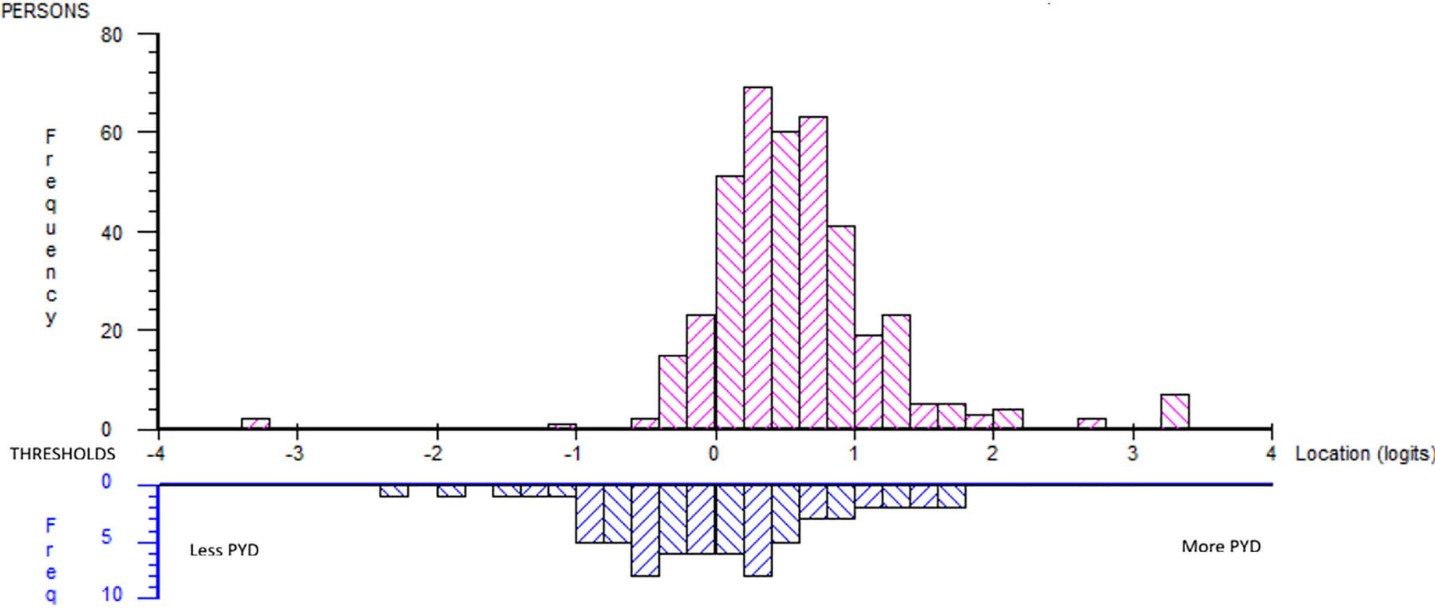

**Fig 3. Person-item distribution five-dimensional subtest analysis.**

Table 6 summarises the item-fit statistics, where the subtest of *Connection* indicates local dependency, but no subtest showed signs of multidimensionality. Yen's Q3* indicated no high residual correlations between subtest pairs.

The summary item-trait interaction was significant ($\chi2 = 55.61$, $p < 0.00$), indicating an overall misfit to the Rasch model. Cronbach's alpha for the subscale was 0.884

In terms of person fit, 6% of the sample's responses did not match the expected Rasch model.

Evidence of multidimensionality was observed in the subtest analysis, as indicated by the relatively high contingency index ($C = 0.710$) and the moderate correlation between subtests ($r = 0.665$). These results suggest that the scale based on five related constructs may not be strictly unidimensional. However, the proportion of non-error variance attributable to the common dimension was very high ($A = 0.923$). This indicates that despite some evidence of multidimensionality, the scale largely reflects a strong underlying and common construct. Therefore, the use of a single composite score can be considered justifiable based on these results.

To add perspective to the question of unidimensionality, the PCA/t-test protocol was implemented, resulting in 6.08% significant t-tests (Agresti-Coull 95% CI 4.1%−8.9%). The loadings of the subdimensions in the first principal factor for the t-test were: *Confidence, Competence,* and *Connection* (>+0.3) in one dimension and *Caring* and *Character* (<−0.3) in the second dimension.

The estimated indices (C, r, and A) did not indicate strict unidimensionality, and the PCA/t-tests results were slightly above 5% (although the 95% CI included 5%). The five subscale analyses showed recurrent significant item-trait interactions, low PSI values, and some person misfit. Therefore, a decision was made to perform a post hoc exploratory subtest analysis of the two potential dimensions suggested by the PCA for comparison.

### Step three: Post hoc two-dimensional subscales and subtest analysis

The first dimension, consisting of *Competence*, *Confidence,* and *Connection* (loaded >+0.3 in the first principal factor), was labelled "*Self-worth*" as the underlying constructs' items are related to the person's experience of their own value from both an intra- and an interpersonal perspective. The second dimension, consisting of *Character* and *Caring* (loaded <−0.3 in the first principal factor)*,* was labelled "*Pro-social*" as the underlying constructs' items are related to the person's experience of their own judgement of my moral and compassionate behaviours in relation to others.

### Self-worth subscale analysis

A lack of fit was reflected in the person–item distribution (logits: mean 0.944, SD 1.217), indicating a mismatch between item difficulty and respondents' ability (Fig 4).

The analysis showed disordered thresholds for i1, i2, i14, i15, i16, and i17. The response "disagree" did not function as expected for these items.

The summary item-trait interaction was significant ($\chi2 = 166.22$, $p < 0.00$), indicating an overall misfit to the Rasch model.

**Table 6. Item-fit statistics subtest analysis of the five dimensions in location order.**

| Testlet | Location | SE | FitResid[a] | χ2 | Prob |
|---|---|---|---|---|---|
| Character | −0.246 | 0.026 | 0.296 | 3.619 | 0.605505 |
| Connection | −0.046 | 0.024 | −4.026 | 27.365 | 0.000048 |
| Caring | −0.045 | 0.026 | 1.585 | 18.110 | 0.002812 |
| Competence | 0.087 | 0.028 | 0.165 | 2.356 | 0.797943 |
| Confidence | 0.251 | 0.023 | 1.110 | 4.158 | 0.526950 |

[a]Fit Residual (FitRes) above 2.5 indicating multidimensionality. FitRes below −2.5 indicating local dependency.

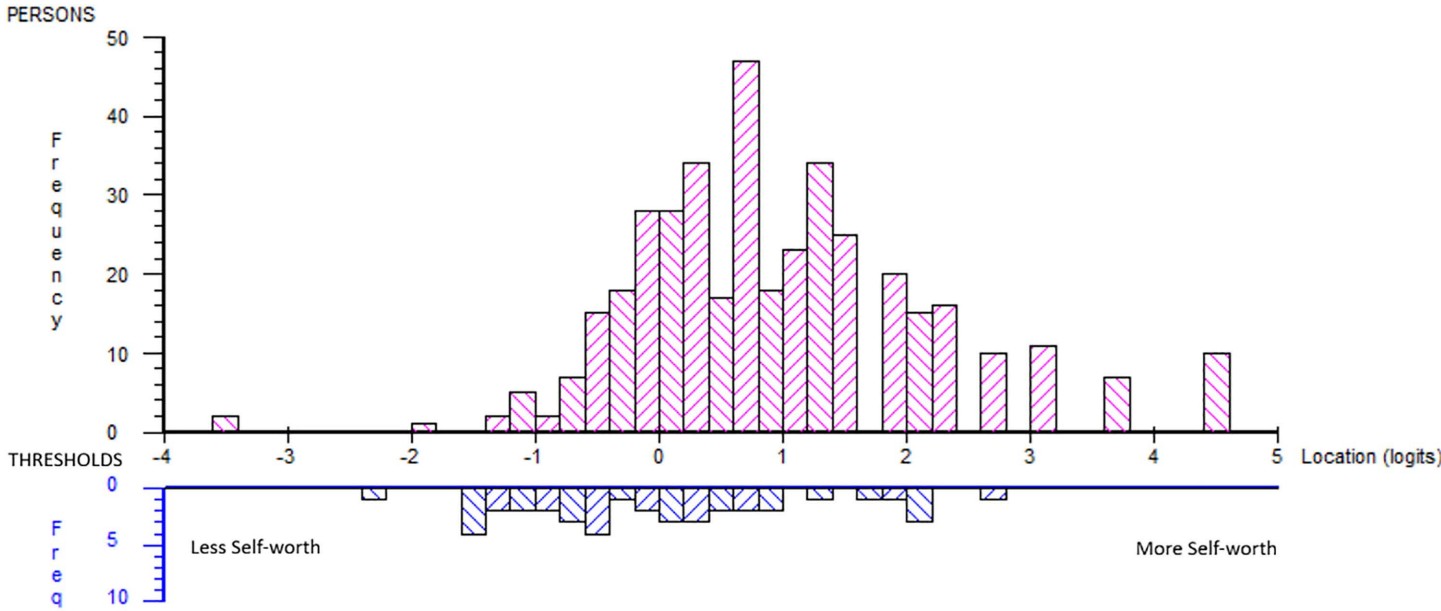

**Fig 4. Person-item distribution post hoc Self-worth subscale.**

Table 7 presents the item-fit statistics in location order. The fit residuals of i2 and i3 indicated multidimensionality, and i7 indicated local dependency in the Self-worth subscale. Yen's Q3* indicated four item pairs with high residual correlations. The correlation pairs followed the underlying structure of the original subscales of *Competence* (three pairs) and *Connection* (one pair).

The PSI for the scale was 0.850. In terms of person fit, 9% of the sample's responses did not match the expected Rasch model.

There was significant (after Bonferroni adjustment) uniform DIF for gender in the subscale Self-worth, in i2 (F-ratio 14.921, girls rated > boys), i4 (F-ratio 21.277, boys > girls), i7 (F-ratio 14.305, boys > girls), and i15 (F-ratio 10.881, girls > boys). After adjusting for DIF for i4, the DIF remained for i7 (F-ratio 18.831, boys > girls) and for i2 (11.525, girls > boys), and disappeared for i15 (artificial DIF). After adjusting for DIF for i7, DIF remained developed in i3 (F-ratio 13.036, boys > girls) and i6 (F-ratio 10.149, boys > girls). After adjusting DIF also for i3, DIF for i6 remained (F-ratio 16.245, boys > girls). After also splitting i6, no DIF remained. In summary, DIF adjustment was needed for i3, i4, i7, and i6, i.e., four out of 10 items.

To assess the clinical significance of the observed DIF for the two subscales, we compared the total person locations between gender groups before and after adjusting for DIF in the respective subscales using the open-source statistical software Jamovi (version 2.2.2), [49]. Thus, DIF adjustment was performed by splitting items, as described above, into separate items for gender. Before adjustment, the mean difference in Self-worth person locations was 0.586 logits (p < .001), with an effect size (Cohen's d) of 0.613 (95% CI: −0.822 to −0.404). After DIF adjustment, the mean difference was −0.423 logits (p < .001), with an effect size of −0.350 (95% CI: −0.554 to 0.146). The overlapping 95% confidence intervals for the effect sizes suggest that DIF does not substantially impact group-level interpretations of Self-worth scores. As six out of ten items showed disordered thresholds, an analysis was performed, rescoring "strongly disagree" and "agree" into one response category for the items with disordered thresholds, without adjustment for DIF (as described above). This resolved all disordered thresholds. The item misfit of i2, i3, and i7 remained, the item-trait interaction was still significant (PSI was rather unchanged at 0.847), and the person misfit within the sample decreased to 8.3%. The DIF

**Table 7. Item-fit statistics for the post hoc two-dimensional subscale analyses. Items presented in location order.**

| Subscale | Item (i) | Location | SE | FitResid[a,b] | χ2 | Prob |
|----------|----------|----------|-----|--------------|-----|------|
| **Self-worth** | | | | | | |
| | i15 | −0.700 | 0.071 | −2.414 | 12.624 | 0.027167[b] |
| | i17 | −0.417 | 0.063 | 0.455 | 2.233 | 0.81611 |
| | i16 | −0.343 | 0.067 | −2.412 | 17.586 | 0.003514 |
| | i1 | −0.244 | 0.061 | 0.223 | 4.415 | 0.491403 |
| | i7 | −0.159 | 0.064 | −3.744 | 26.677 | 0.000066 |
| | i2 | 0.042 | 0.065 | 2.876 | 18.606 | 0.002277 |
| | i14 | 0.255 | 0.069 | 0.438 | 2.832 | 0.72594 |
| | i4 | 0.273 | 0.060 | −1.594 | 15.144 | 0.009766[b] |
| | i6 | 0.610 | 0.057 | −0.692 | 3.305 | 0.653046 |
| | i3 | 0.683 | 0.054 | 6.367 | 62.797 | 0.000000 |
| **Pro-social** | | | | | | |
| | i9 | −0.393 | 0.077 | 0.511 | 2.624 | 0.757696 |
| | i11 | −0.359 | 0.074 | −2.755 | 32.76 | 0.000004 |
| | i12 | −0.267 | 0.070 | −3.039 | 24.972 | 0.000143 |
| | i13 | −0.243 | 0.068 | −2.924 | 22.699 | 0.000386 |
| | i8 | 0.133 | 0.063 | −0.153 | 10.922 | 0.052943 |
| | i10 | 0.329 | 0.059 | 0.958 | 2.393 | 0.792467 |
| | i5 | 0.800 | 0.054 | 7.072 | 88.438 | 0.000000 |

a Fit Residual (FitRes) above 2.5 indicating multidimensionality. FitRes below −2.5 indicating local dependency

b Not significant after Bonferroni adjustment.

pattern for gender in i3, i4, i14, and i15 was still present, and i16 now presented a DIF in which girls consistently scored higher than boys.

### Pro-social subscale analysis

A lack of fit was reflected in the person–item distribution, indicating a mismatch between item difficulty and the respondents' ability (Fig 5). This indicates that the scale failed to represent higher levels of experienced Pro-social behaviours where a portion of the sample was located (logits: mean 1.234, SD 1.104).

The analysis showed disordered thresholds for i8, i9, i10, i11, i12 and i13. For i8, i10, i12 and i13, the response category "disagree" did not function as expected. For i9 and i11, both the response categories "strongly disagree" and "disagree" did not function as expected.

The summary item-trait interaction was significant ($\chi2 = 184.81$, $p < 0.00$), indicating an overall misfit to the Rasch model.

The item-fit statistics in Table 7 indicate multidimensionality in i5 and i11, i12 and i13 indicated local dependency. Yen's Q3* indicated four high residual correlations between item pairs. The correlation pairs mainly followed the underlying structure of the original subscale *Caring*.

The PSI for the scale was 0.715 In terms of person fit, 13% of the sample's responses did not match the expected Rasch model. There was significant (after Bonferroni adjustment) DIF for gender in the subscale Pro-Social, for i12 (F-ratio 18.293, girls rated > boys), i13 (F-ratio 14.848, girls > boys), and i11 (F-ratio 10.153, girls > boys). After adjusting for DIF for i12, the DIF for i13 remained (F-ratio 18.786, girls > boys) and also for i11 (F-ratio 13.542, girls > boys). After

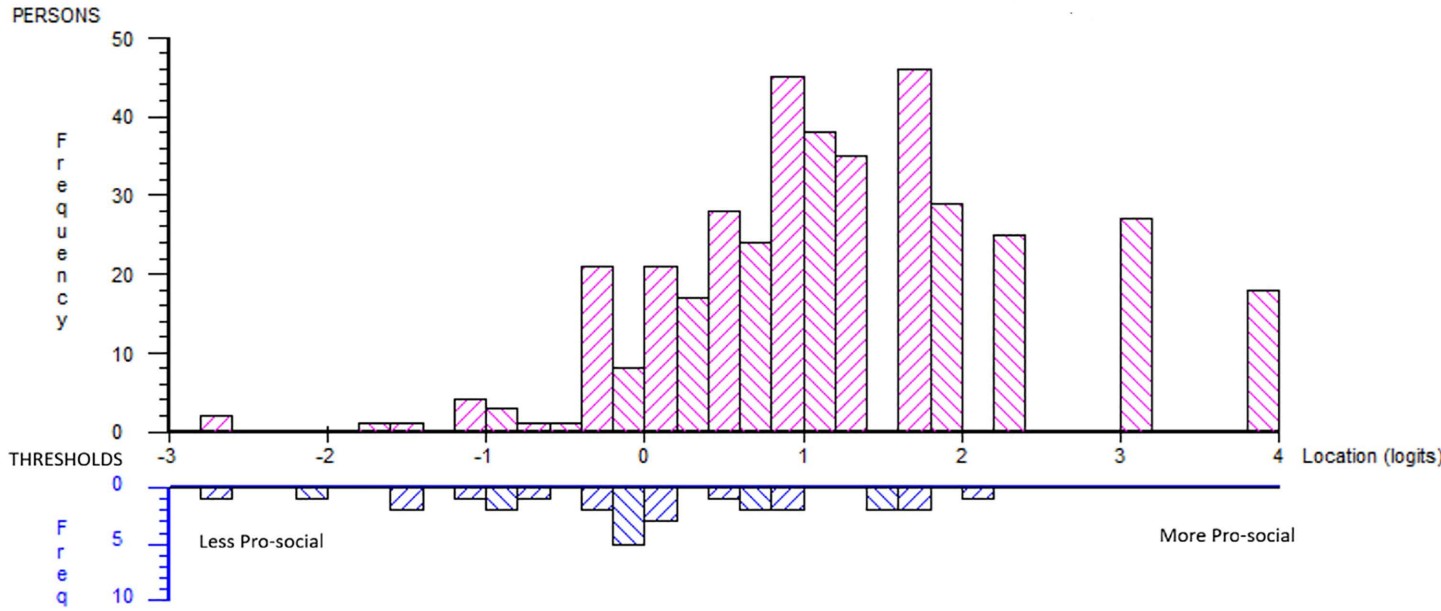

**Fig 5. Person-item distribution post hoc Pro-social subscale.**

also adjusting for DIF in i13, the DIF for i11 remained (F-ratio 18.156, girls>boys). After adjusting for DIF in i11, no DIF remained. In summary, DIF adjustment was needed for i11, i12 and i13, i.e., for three out of seven items.

Before adjustment, the mean difference in Pro-Social person locations was 0.096 logits (p = 0.303), with an effect size (Cohen's d) of 0.106 (95% CI: −0.096 to 0.308). After adjustment, the mean difference was −0.102 logits (p = 0.367), with an effect size of −0.093 (95% CI: −0.295 to 0.109). The overlapping 95% confidence intervals for the effect sizes suggest that DIF does not substantially impact group-level interpretations of BAS-2 scores.

As six out of seven items showed disordered thresholds, an analysis was performed, rescoring "strongly disagree" and "disagree" to one category, without adjustment for DIF (as described above). The rescoring resolved all but the i5 disordered thresholds. The item misfit pattern was the same, item-trait interaction ($\chi$ = 188.39, p < 0.00) was still significant, and PSI was stable at 0.732. Item 5 still indicated multidimensionality, and i11, i12 and i13 still indicated local dependency. However, i12 and i13 now indicated a gender DIF where girls consistently reported slightly higher scores than boys.

### Two-dimensional subtest analyses

Since DIF did not substantially impact group-level interpretations in either of the two scales, we continued the two-dimensional analysis without DIF adjustment. The subtest analysis showed model fit in the person–item distribution, indicating a match between item difficulty and the respondents' ability (Fig 6). The scale succeeded in representing most of the sample (logit: mean 0.396, SD 0.428) with only a minor gap at the higher end of the continuum.

The summary item-trait interaction was not significant ($\chi$2 = 14.60, p = 0.15), also indicating fit to the Rasch model.

Table 8 summarises the item-fit statistics, presenting the two superitems in location order (The item statistics are based upon the initial item statistics; hence, without rescored thresholds or adjustment for DIF). There were no signs of multidimensionality or local dependency. Yen's Q3* indicated no high residual correlation between the two subtests.

Cronbach's alpha for the scale was 0.884.

In terms of person fit, 3% of the sample's responses did not match the expected Rasch model.

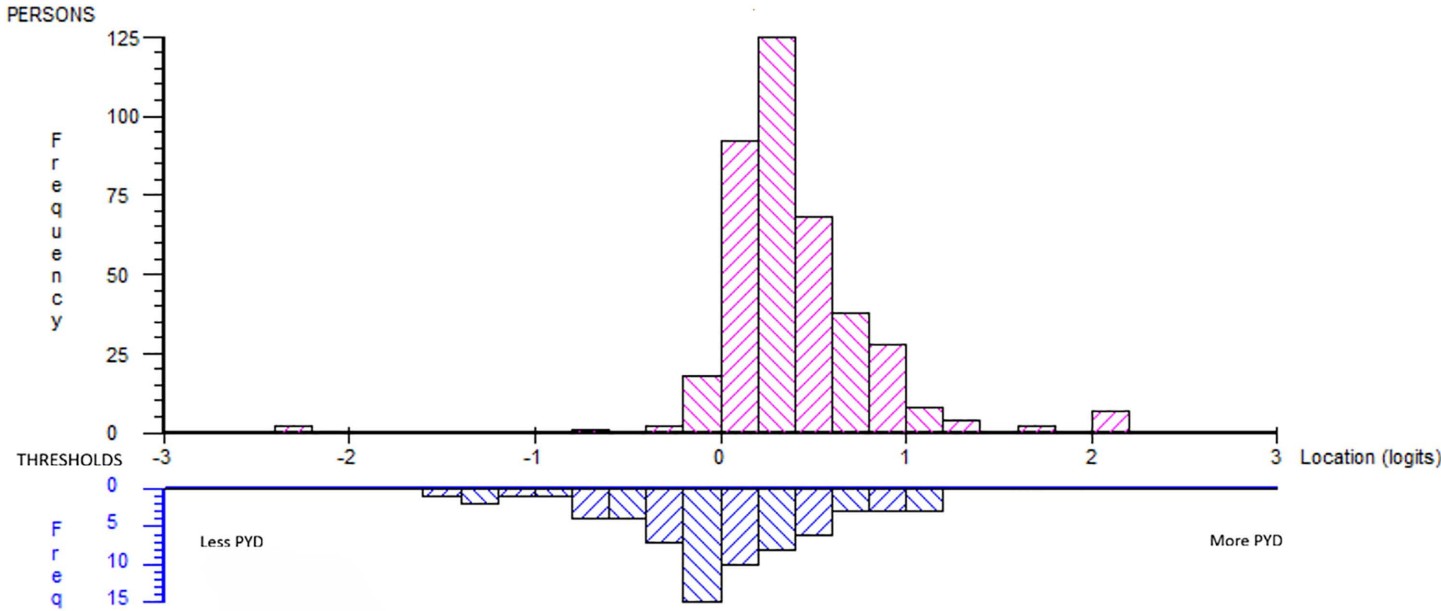

**Fig 6. Person-item distribution post hoc two-dimensional subtest analysis.**

**Table 8. Item statistics for the post hoc two-dimensional subtest analyses in location order.**

| Testlet | Location | SE | FitResid[a] | χ2 | Prob |
|---|---|---|---|---|---|
| Self-worth | −0,004 | 0,011 | −2,279 | 9,5 | 0,090723 |
| Pro-social | 0,004 | 0,014 | 1,298 | 5,099 | 0,403923 |

[a]Fit Residual (FitRes) above 2.5 indicating multidimensionality. FitRes below −2.5 indicating local dependency.

In the subtest analysis, the contingency index (C = 0.656) and the moderate correlation between subtests (r = 0.699) indicated minor evidence of multidimensionality. The proportion of common non-error variance (A = 0.801) was lower than in the previous subtest analysis of the five-dimensional subtest, but still suggested that a substantial part of the reliable variance could be attributed to a common dimension. Thus, the unidimensionality remained acceptable in this sample.

Unidimensionality was also supported by the PCA/t-test protocol, where the Self-worth dimension (>+0.3 loading in the first principal factor) was tested against the Pro-social dimension (<−0.3 loading in the first principal factor), resulting in 1.52% significant t-tests (Agresti-Coull 95% CI 0.03%−3.4%).

### Score transformation for the two subscales

Since DIF did not substantially impact group-level interpretations in either of the two scales, score transformation was carried out without correcting for DIF (S2 Appendix). Neither were disordered thresholds considered in the transformation process. In the appendix, the two subscale scores, for Self-worth and Pro-social, were converted into linear logit values, accompanied by their standard errors, and further transformed into linearised scores using an online conversion tool [48].

### Discussion

The RMA of the present sample indicated that the original 17-item PYD-VSF did not fit the Rasch model, demonstrated multidimensionality, and showed multiple signs of local dependency between item pairs within the suggested five

subscales. This may have caused an overrated reliability of the overall scale (PSI 0.874). It should also be noted that applying a unidimensional model to multidimensional data can create local dependencies [42].

As the 5C subscales' individual items were added to subscales in the subtest analysis, the local dependencies from the 17-item scale analysis were absorbed, resulting in a slight increase in reliability (alpha 0.884). This indicates that items might be very closely related and are better treated as subtests/superitems than individual items to represent the continuum of PYD from low to high. The precision and targeting were improved, and no residual correlations above 0.2 appeared in this analysis. This strengthens the theory of *Competence, Confidence, Character, Caring* and *Connection* being underlying constructs of PYD.

When the multidimensionality was tested in the five-dimensional subtests, there were indications from the PCA that there might be an underlying two-dimensional structure. As the post hoc two-dimensional structure was explored, it generally showed a better fit to the expected Rasch model, with high reliability (alpha 0.884). Furthermore, it also showed better targeting and precision, as well as a lower percentage of significant tests (PCA/t-test) of person estimates: 1.52% compared to 6.08% for the five-dimensional subtest analyses. If only the subtest indices of C, r and A were considered, both the five-dimensional and two-dimensional structures could be argued to show minor indication of multidimensionality, but adding the PCA/t-test results provides additional knowledge, and a more justifiable decision to accept the two-dimensional solution can be made [21]. Anyhow, there are still arguments for using the PYD-VSF as two separate scales, rather than as one scale.

From a practical standpoint, a two-dimensional solution appears particularly advantageous. While a five-dimensional model may provide greater theoretical nuance, it also increases the complexity of both measurement and interpretation, which may limit its applicability in real-life settings. In contrast, a two-dimensional scale allows for a more straightforward administration, scoring, and communication of results, making it more feasible for use in applied contexts such as schools, youth programmes, or clinical practice. Importantly, a more concise structure does not necessarily imply a loss of substantive meaning; rather, it may serve to highlight the most essential components of PYD in ways that are both psychometrically sound and practically meaningful. Thus, the two-dimensional model not only represents an analytically robust alternative but also holds promise for bridging the gap between theoretical advancements in PYD research and their translation into accessible, user-friendly tools for practitioners [25].

The subtest analysis of the post hoc two-dimensional solution indicated a relatively good fit to the Rasch model (no significant item-trait interaction), good reliability (α 0.884) and targeting (logits: mean 0.396, SD 0.428), and only 3% of the sample's responses did not match the expected Rasch model. Furthermore, the location order indicated that Self-worth seems to be a prerequisite for Pro-social; hence. the two dimensions are related constructs within the overall PYD. The relationship between underlying constructs has previously been illustrated using a metaphor of several small ropes intertwined into a thicker, more robust one [50]. In this context, Self-worth and Pro-social can be seen as distinct yet interrelated strands that together contribute to the overall construct of PYD. However, since Self-worth seems to be a prerequisite for Pro-social, it seems meaningful to treat the two dimensions separately rather than to combine them into a single total score.

An interesting result is the gender DIF, where boys generally scored higher on items belonging to the Self-worth dimension and girls higher on items belonging to the Pro-social dimension. These findings are important to explore further in future studies [47]. Previous studies based on CTT have not identified similar DIF patterns in total scores [3,23]. However, CTT analysis has found DIF in gender on the individual five subscales [22,51]. DIF in different directions for boys and girls might even cancel each other out when items are summed up to total scores. Importantly, the DIF did not substantially impact group-level interpretations in any of the two scales in this study. It should be noticed that the previous CTT analyses were primarily performed on group level on total score, based on ordinal data and correlations. RMA, on the other hand, is based on item statistics and the probability of endorsing an answer after ordinal data has been transformed to linear logits [42,50]; hence, more nuance to the discussion of possible DIF is brought to the table

with RMA. In our analyses, however, the DIF-adjusted scales did not differ in any significant way from the non-adjusted scales. Therefore, we suggest that the PYD-VSF can be used for both boys and girls without the need for separate scoring. Nevertheless, some gender-related differences in responses could be expected, as these may mirror social and cultural norms. Such factors should be taken into account when evaluating DIF, while recognising that they do not necessarily warrant statistical adjustment [47].

Table 2 indicates that most of the sample was located on the agreement spectrum. Given the initial dysfunction observed in the response categories, a revision of item wording, response options, or both may be necessary to more accurately capture the intended continuum. Although the Likert scale from "strongly disagree" to "strongly agree" is commonly used, the disagreement spectrum tended not to fully function as expected; hence, signs of response order effects or social desirability bias may be present in the results [52]. To strengthen the validity of the measure, it could be argued that new cognitive interviews focusing on this issue could be recommended before further psychometric testing.

As Lerner and Chase [20] address the need to improve the PYD instruments (original, short, and very short versions) for optimising the possibility to capture change over time, it is important to emphasise methodology that can create reliable linear measurements. Considering that the two subscales had a "good-enough" compliance with the Rasch Measurement Model, we decided to make score transformations for the two subscales separately in this study. This allows for more precise analyses of developmental trajectories, beyond the limitations of raw ordinal scores, in other words, inferences regarding the magnitude of change, differences, and calculations of means [48].

The consolidated two-dimensional solution found in this study resonates with the concepts of agency and resilience, which are the least explored but most robust explanations for younger persons' outcomes later in life [53]. Self-worth (*competence, confidence, connection*) embodies an agentic foundation, reflecting young persons' perceived capacities, efficacy, and relational embeddedness that together enable intentional choice and purposive action [8,53]. This resonates with the conceptualisation of agency as the ability to make choices and act upon them within enabling or constraining opportunity structures. Pro-Social (*caring, character*) reflects a moral-relational orientation that facilitates empathy, pro-social behaviour, and adherence to shared norms, thereby potentially activating protective mechanisms under conditions of stress and adversity [53,54]. Furthermore, the hierarchy (Self-worth being a potential requisite for Pro-social) illustrates how agency and resilience are closely linked and the fact that resilience is a very complex phenomenon relying on multiple levels of personal functioning [54]. Hence, the post hoc two-dimensional solution aligns with theoretical models of agency and resilience, understood not as fixed individual traits but as a set of dynamic processes that buffer risk exposure and foster adaptive recovery [54]. From this perspective, Self-worth could be argued to primarily nurture agency, while Pro-social primarily strengthens resilience, yet both operate synergistically to become that thicker rope where mastery experiences and social validation reinforce one another across developmental years. Taken together, the post hoc two-dimensional solution offers easier administration while still cohering with the complex life-course and socio-ecological perspectives, emphasising that young persons' developmental assets relate to multiple domains and stages of life [8].

To summarise, we see a need for more work under the PYD framework, prioritising standardised and validated instruments to measure constructs under the framework to allow further investigation of how these variables might both strengthen the person here and now while also buffering against future adverse outcomes.

## Strengths and limitations

The major strength of this study is the robust RMA that has been described in detail. It provides an insight into the need to consider more than one test to fully understand the validity and reliability of an instrument to operate at a personal level. Another strength of the study is that the target group was involved in the cultural adaptation and translation process of the instrument. Engaging the target group can provide perspectives affecting user-friendliness, which in the long term will influence the data that is collected and analysed, thus also affecting the statistical inferences.

One important limitation is the use of the shortest instrument, the PYD-VSF. If the more extensive version, the PYD-SF, consisting of 36 items, or the full version with 78 items, had been used, it might have provided different results in terms of targeting and Rasch model fit. As item reduction took place, aspects of the different subscales or PYD might have been lost – aspects that might have contributed to a better understanding of the hierarchical order, or that could have targeted the higher levels of the scale where items tended to be missing. Furthermore, the study has a cross-sectional design and data collection took place in the southern region of Sweden. Moreover, data was collected in cooperation with upper secondary schools; hence, students not attending these schools are not represented in the study. Another limitation could be the age group in the sample, as most PYD literature relates to younger samples, but as Dvorsky, Kofler [23] argue, it is important to strive for a tool that can be used across different ages, and the age group of 15–18 needs to be more repre-sented in the future.

## Conclusion

In conclusion, the present study suggests that the PYD-VSF is best represented as a two-dimensional measure, dis-tinguishing between Self-worth and Pro-social. This structure demonstrated good psychometric properties, enhanced interpretability, and practical feasibility compared to both the unidimensional and five-dimensional alternatives. While gender-related DIF was observed, it did not substantially affect overall interpretations and may reflect social and cultural response tendencies rather than measurement bias. Importantly, the two-dimensional solution aligns with theoretical perspectives on agency and resilience, underscoring its potential to capture meaningful developmental processes. Taken together, these findings support the use of the PYD-VSF as two separate yet interrelated scales that can contribute to both research and applied practice in promoting positive youth development. Future psychometric research should focus on further testing and validating the two-dimensional solution as well as longitudinal analyses to ensure both measure-ment invariance and sensitivity to change over time. This may strengthen the instrument's ability to capture complex developmental processes and thereby contribute to more robust conclusions within arenas where PYD initiatives are common, such as schools or sports.

## Supporting information

**S1 Appendix. Person-Item distribution of the five separate subscales.**
(PDF)

**S2 Appendix. Linear transformed scores**
(DOCX)

**S1 File. Striking image** .
(TIF)

## Acknowledgments

We sincerely thank all the participants in the cross-sectional study and the panel contributing to our translation process. A special thank you goes to Master student Jacquline Söderström for valuable discussions during the initial explorative RMA of the PYD-VSF.

## Author contributions

**Conceptualization:** Johanna Bergman, Maria Haak, Åsa Bringsén, Petra Nilsson Lindström, Albert Westergren.

**Data curation:** Johanna Bergman.

**Formal analysis:** Johanna Bergman, Albert Westergren.

**Investigation:** Johanna Bergman, Albert Westergren.

**Methodology:** Johanna Bergman, Albert Westergren.

**Project administration:** Johanna Bergman, Maria Haak.

**Supervision:** Maria Haak.

**Validation:** Johanna Bergman, Albert Westergren.

**Writing – original draft:** Johanna Bergman.

**Writing – review & editing:** Johanna Bergman, Maria Haak, Åsa Bringsén, Petra Nilsson Lindström, Albert Westergren.

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
