## [Decision Letter · Decision Letter 0]

2 Aug 2025

Dear Dr. Bergman,

Thank you for submitting your manuscript to PLOS ONE. After careful consideration, we feel that it has merit but does not fully meet PLOS ONE’s publication criteria as it currently stands. Therefore, we invite you to submit a revised version of the manuscript that addresses the points raised during the review process.

We look forward to receiving your revised manuscript.

Kind regards,

Karl Bang Christensen, Ph.D.

Academic Editor

PLOS ONE

**Journal Requirements:**

1. When submitting your revision, we need you to address these additional requirements. Please ensure that your manuscript meets PLOS ONE's style requirements, including those for file naming. The PLOS ONE style templates can be found at https://journals.plos.org/plosone/s/file?id=wjVg/PLOSOne_formatting_sample_main_body.pdf and https://journals.plos.org/plosone/s/file?id=ba62/PLOSOne_formatting_sample_title_authors_affiliations.pdf 2. In the ethics statement in the Methods, you have specified that verbal consent was obtained. Please provide additional details regarding how this consent was documented and witnessed, and state whether this was approved by the IRB. 3. Thank you for stating the following financial disclosure: Funded by Kristianstad University  Please state what role the funders took in the study.  If the funders had no role, please state: "The funders had no role in study design, data collection and analysis, decision to publish, or preparation of the manuscript." If this statement is not correct you must amend it as needed. Please include this amended Role of Funder statement in your cover letter; we will change the online submission form on your behalf. 4. Thank you for stating the following in the Acknowledgments Section of your manuscript: We sincerely thank all the participants in the cross-sectional study and the panel contributing to our translation process. A special thank you goes to Master student Jacquline Söderström for valuable discussions during the initial explorative RMA of the PYD-VSF. The study was conducted in line with the WMA Declaration of Helsinki (WMA, 1964) and approved by the Swedish Ethical Review Authority Dnr 2024-01088-01. The study is part of a PhD project funded by Kristianstad University. We note that you have provided funding information that is not currently declared in your Funding Statement. However, funding information should not appear in the Acknowledgments section or other areas of your manuscript. We will only publish funding information present in the Funding Statement section of the online submission form. Please remove any funding-related text from the manuscript and let us know how you would like to update your Funding Statement. Currently, your Funding Statement reads as follows: Funded by Kristianstad University Please include your amended statements within your cover letter; we will change the online submission form on your behalf. 5. Your ethics statement should only appear in the Methods section of your manuscript. If your ethics statement is written in any section besides the Methods, please delete it from any other section. 6. We note that this data set consists of interview transcripts. Can you please confirm that all participants gave consent for interview transcript to be published? If they DID provide consent for these transcripts to be published, please also confirm that the transcripts do not contain any potentially identifying information (or let us know if the participants consented to having their personal details published and made publicly available). We consider the following details to be identifying information:- Names, nicknames, and initials- Age more specific than round numbers- GPS coordinates, physical addresses, IP addresses, email addresses- Information in small sample sizes (e.g. 40 students from X class in X year at X university)- Specific dates (e.g. visit dates, interview dates)- ID numbers Or, if the participants DID NOT provide consent for these transcripts to be published:- Provide a de-identified version of the data or excerpts of interview responses- Provide information regarding how these transcripts can be accessed by researchers who meet the criteria for access to confidential data, including:a) the grounds for restrictionb) the name of the ethics committee, Institutional Review Board, or third-party organization that is imposing sharing restrictions on the datac) a non-author, institutional point of contact that is able to field data access queries, in the interest of maintaining long-term data accessibility.d) Any relevant data set names, URLs, DOIs, etc. that an independent researcher would need in order to request your minimal data set. For further information on sharing data that contains sensitive participant information, please see: https://journals.plos.org/plosone/s/data-availability#loc-human-research-participant-data-and-other-sensitive-data If there are ethical, legal, or third-party restrictions upon your dataset, you must provide all of the following details (https://journals.plos.org/plosone/s/data-availability#loc-acceptable-data-access-restrictions):a) A complete description of the datasetb) The nature of the restrictions upon the data (ethical, legal, or owned by a third party) and the reasoning behind themc) The full name of the body imposing the restrictions upon your dataset (ethics committee, institution, data access committee, etc)d) If the data are owned by a third party, confirmation of whether the authors received any special privileges in accessing the data that other researchers would not havee) Direct, non-author contact information (preferably email) for the body imposing the restrictions upon the data, to which data access requests can be sent 7. If the reviewer comments include a recommendation to cite specific previously published works, please review and evaluate these publications to determine whether they are relevant and should be cited. There is no requirement to cite these works unless the editor has indicated otherwise. 

Reviewers' comments:

**Comments to the Author**

1. Is the manuscript technically sound, and do the data support the conclusions?

Reviewer #1: Partly

Reviewer #2: No

2. Has the statistical analysis been performed appropriately and rigorously?

Reviewer #1: No

Reviewer #2: No

3. Have the authors made all data underlying the findings in their manuscript fully available?

Reviewer #1: No

Reviewer #2: Yes

4. Is the manuscript presented in an intelligible fashion and written in standard English?

Reviewer #1: Yes

Reviewer #2: Yes

**Reviewer #1:**  POND-D-25- 20352

"Exploring the dimensionality of the Positive Youth Development 5C Very Short Form using Rasch Measurement Theory in Swedish Upper Secondary School Contexts.".

General Introduction

The manuscript explores the background to Positive Youth Development (PYD) where the most commonly used model was that with the five dimensions of Competence, Confidence, Character, Caring and Connection. To this end a PYD 5C questionnaire was developed that has been used internationally to measure youth development. The original PYD 5C instrument consisted of 78 items (13), with a 38-item short form (PYD-SF) and a 17-item very short form (PYD-VSF) created to enhance practical usability. It is this latter version which is the subject of the reported study. Each dimension is a subscale consisting of three or four items that are argued to function on their own using an average of included items or adding up all 17 items and calculating a mean to represent the overall total. Each item is scored on a 5-point Likert scale.

Methods

The study comprised two elements; a) an adaptation of the questionnaire into Swedish using a dual panel approach with cognitive debriefing and b) a psychometric analysis based upon Rasch Measurement Theory.

148-149. Note that these statements are targeted at the fit statistics in the RUMM2030 programme, and is not the case if, for example, conditional chi-square fit is used.

162 It might be helpful to the readership if a more structured description of methodology is provided. Given the way in which the scale is used, one would initially expect results from the summation of the 17 items, and then for each separate dimension, or vice-versa. Consequently, some statement along the lines of “For the 17 item scale and each subscale the following aspects of Rasch model compliance were examined”.

Item Monotonicity

Initially ( but not subsequently when testlets are used) what were the threshold patterns of the items.

Local Item dependency (LID)

Definition. What was the extent and pattern of LID if present. If so, how did this relate to the five dimensions

Item & Person Fit

Definition. The Chi-square fit and item residuals, reported at the item level. Person fit reported at a group level if misfit present. Summary item-trait Chi-Square.

164. Why if the Chi-Square fit not considered here? This is the fit most often reported, and the comment about sample size is based upon the Type 1 error rate of the Chi-Square fit.

Differential Item Functioning (DIF)

Definition. What contextual factors are chosen to examine DIF

Unidimensionality

Definition. Method and parameters to identify multi-dimensionality

Approaches to resolve problems when apparent (or alternatively the options could be specified within each of the sections above.

What options are available to resolve any problems found. For example, rescoring, testlet construction. The method for each option should be described. It should also be noted that in the context of any two-testlet result, a conditional chi-square test of fit is provided.

190. It is not clear as to whether or not this strategy was a result of the findings of the analysis to-date, i.e. item-level, or determined a-priori. If the former, then it is a result, not method, implying that multidimensionality had been found. If the latter, then should be clearly explained in the options section, but without necessarily pre-judging which item sets will be grouped.

195. The authors may like to review their interpretation of C in the context of Andrich’s example in the paper:

“Table 2 also shows that because of a relatively large value for c (0.722), the summary latent correlation between different subscales is a relatively low 0.658. Nevertheless, A α (the proportion of the scale common variance relative to the common and unique variance, that is, the proportion of the non-error variance), is a relatively high 0.9 (0.886). This indicates that the subscales were, in general, correlated sufficiently highly that, together with the four subscales, the greatest component of variance is the true, common variance. …….. This high proportion of true common variance of the scale suggests that for the purposes to which ASAT was put, using a single score was generally justifiable.

Thus they need to review their findings and the implications for the analysis that follows.

200. There is some risk here that any observed multidimensionality is simply the result of local dependency. Indeed, at this stage it is becoming unclear as to which item sets are grouped. Does this analysis follow from the 17 item set, or dimension-based testlets.

Results

210. As the items have not yet been introduced they need explicit identification.

222. It would be helpful if it was made clear that the analysis following related to the item level analysis of the 17 item scale

225. A Table reporting chi-square and residual fit statistics for the 17 items should be provided to give the reader the staring point for the analysis. Also the summary item-trait interaction Chi-Square and reliability should be reported here, and also for each subsequent analysis revision.

231. These three items belong to the same subscale, indicating potential LD

234. Removal of items should normally be a last resort but in light of the explanation given, this may be correct in this instance. As they belong to different subscales their removal should not compromise individual subscale analysis. However, it would have been informative if they also showed serious misfit within each subscale, which would have added to the veracity of the decision to remove them. Furthermore, individual item fit may be confounded by the presence of LD and the subscale level analysis may help inform on this.

237. The problem with item 5 should be picked up in the discussion as it reflects on the adaptation process, i.e. double negatives should not have arisen.

244. Is this related to subtest analysis within each subscale, or at the subscale level for the whole scale? For the interpretation of the result, see the comment at 195 above.

254. Give the above (195) the multidimensionality conclusion is not correct, but the residual correlations of the subtest is informative, and the subsequent two domain result

262-263. The two domains, taken, together, have been shown to fit the Rasch model. Essentially the testlets have absorbed the LD and its influence upon individual domain fit. Using the test-equating procedure on RUMM2030 the authors could now produce a transformation table for the total score, and each domain, all on the same metric (reminding readers that two items from the 17 were removed).

The issue then arises, why try and analyze the item set of each domain without resolving any of the problems that arise? This leaves the readers with the conundrum of having seen the two domains emerge in the testlet analysis with apparently good fit etc., but now showing poor fit at the item level! For purposes of revealing the reliability of the new domain, this may be best resolved in future analysis involving a test-retest situation

Discussion

343. Where is the results for item fit of each subscale which supports this conclusion?

378. This implies satisfactory fit of each domain, yet the analytical conclusion is one of misfit .

**Reviewer #2: ** The authors translated and cross-culturally adapted the 5C Positive Youth Development Very Short Form (PYD-VSF) in 430 upper secondary school pupils. Their results showed that the data did not fit the Rasch model. Below my comments

More information should be provided about the translation process. What were the competences of these 6 lay people? How was the translation process carried out? How many reviewers translated the version from English to Swedish and how many people from Swedish to English? What was the composition of the expert panel that evaluated the translations? How was the final Swedish version developed? Report the demographic characteristics of the 15 pupils involved in this process.

The authors decided to delete items 3 and 5 due to poor fit to the Rasch model. Why did the authors make this drastic choice? I do not agree with their choice. First of all, considering that the scale is already published and is used with this structure, I would try to modify the scale without deleting themt, but with the creation of subtests of items that show local dependence to try to avoid eliminating items 3 and 5. If this attempt fails, then I would eliminate the items that misfit the model.

The authors found disordered thresholds in six out of nine items in the Self-worth subscale and no items in the pro-social subscale. How did they solve this problem?

The authors found a uniform DIF by gender for 3 items in the Self-worth subscale. How did they solve this problem?

Overall, it seems that the authors did not adequately address the problems of the scale (disordered thresholds, local dependence, presence of DIF); probably the lack of fit to the model that the authors report is due to these unresolved problems of the scale. I recommend resolving the local dependency (trying not to remove any items), resolving the messy thresholds, and re-evaluating the model fit.

**Do you want your identity to be public for this peer review?** For information about this choice, including consent withdrawal, please see our Privacy Policy

Reviewer #1: No

Reviewer #2: **Yes: ** Leonardo Pellicciari

---

## [Author Response · Author response to Decision Letter 1]

18 Nov 2025

Thank you for the opportunity to submit a revised version of our manuscript (ID: PONE-D-25-20352), entitled Exploring the dimensionality of the positive youth development 5C very short form using Rasch measurement theory in Swedish upper secondary school contexts.

We sincerely appreciate the valuable feedback and are grateful to the reviewers for their insightful and constructive comments. Based on these suggestions, we have made corresponding revisions throughout the manuscript. All changes are highlighted in yellow, except for the Methods, Results and Discussion sections, which have been substantially revised in their entirety, hence the headlines have been highlighted in yellow. We therefore kindly refer to a thorough reading of these sections alongside our detailed responses below.

We look forward to your response and remain happy to address further questions or comments.

Please find below our point-by-point response to reviewer’s comments and concerns.

Reviewer Comment 1:

Line148-149: Note that these statements are targeted at the fit statistics in the RUMM2030 programme, and is not the case if, for example, conditional chi-square fit is used.

Response: Thank you for pointing this out. As the Method section has undergone major revision this is no longer stated and the reference is not used.

Reviewer Comment 2:

Line 162: It might be helpful to the readership if a more structured description of methodology is provided. Given the way in which the scale is used, one would initially expect results from the summation of the 17 items, and then for each separate dimension, or vice-versa. Consequently, some statement along the lines of “For the 17 item scale and each subscale the following aspects of Rasch model compliance were examined”.

Response: Thank you for your valuable comment regarding the structure of the methodology. The Method section has been thoroughly revised and reorganized to provide a clearer and more systematic description of the analytical procedures. This revised structure is consistently reflected in the Result section, enabling the reader to more easily follow the logic and progression of the analysis (Page 8, line 166-185).

Reviewer Comment 3:

Item Monotonicity

Initially (but not subsequently when testlets are used) what were the threshold patterns of the items.

Response: Thank you for committing on this. The Result section is now updated in terms of reporting item monotonicity. In Table 3 on page 14 an overview of thresholds patterns is presented for the initial 17-items, five-dimension and the post hoc two-dimension analyses. The initial thresholds patterns are also described in the text (Page 15, line 228-229)

Reviewer Comment 4:

Local Item dependency (LID)

Definition. What was the extent and pattern of LID if present. If so, how did this relate to the five dimensions.

Response: Thank you for suggesting clarification of LID. The Result section is now updated in terms of reportin LID and Table 3 on page 14 provides an overview of LID patterns for all analyses. The text is also updated to more thoroughly present the initial pattern of LID in the initial 17-item analysis and how they related to the five dimensions (as an example see Page 15, line 233-236).

Reviewer Comment 5:

Item & Person Fit

Definition. The Chi-square fit and item residuals, reported at the item level. Person fit reported at a group level if misfit present. Summary item-trait Chi-Square.

Response: Thank you for your suggestion to enhance the reporting of item and person fit. A clear definition has now been included in Table 1 (page 9). The extensively revised Results section provides a more detailed presentation of item and person fit across all analyses, including summary and item-trait Chi-Square statistics. In addition, Table 3 (page 14) has been added to give an overview of item and person fit for the different analyses.

Reviewer Comment 6:

Line 164: Why if the Chi-Square fit not considered here? This is the fit most often reported, and the comment about sample size is based upon the Type 1 error rate of the Chi-Square fit.

Response: Thank you for commenting on this. The Chi-square fit is now reported for all the analyses and is reported in the major revisions of the Results section.

Reviewer Comment 7:

Differential Item Functioning (DIF) Definition. What contextual factors are chosen to examine DIF

Response: Thank you for your suggestion to clarify DIF. The updated method section and Table 1 (page 9) has been added to clarify the definition and the chosen factors of school year and gender are presented in the Table 1. Table 3 (page 14) has also been added to provide an overview of the DIF results from the different analyses.

Reviewer Comments 8 & 9:

Unidimensionality

Definition. Method and parameters to identify multi-dimensionality

Approaches to resolve problems when apparent (or alternatively the options could be specified within each of the sections above.

What options are available to resolve any problems found. For example, rescoring, testlet construction. The method for each option should be described. It should also be noted that in the context of any two-testlet result, a conditional chi-square test of fit is provided.

Line 190: It is not clear as to whether or not this strategy was a result of the findings of the analysis to-date, i.e. item-level, or determined a-priori. If the former, then it is a result, not method, implying that multidimensionality had been found. If the latter, then should be clearly explained in the options section, but without necessarily pre-judging which item sets will be grouped.

Response: Thank you for your suggestion to clarify what methods and parameters used to identify multidimensionality. Table 1 (page 9) has been added to both provide both definitions and interpretations applied in this process. This extensively revised Method section has also been updated to describe how potential issues were addressed. Any problems identified during the analyses were managed accordingly, and rescoring was performed when most items showed disordered thresholds. Furthermore, the two-dimensional model is now explicitly described as a post-hoc procedure, as it resulted from the findings of the initial analyses (Page 8, line 180-185 and Page 8, line 172-175).

Reviewer Comment 10:

195. The authors may like to review their interpretation of C in the context of Andrich’s example in the paper:

“Table 2 also shows that because of a relatively large value for c (0.722), the summary latent correlation between different subscales is a relatively low 0.658. Nevertheless, A α (the proportion of the scale common variance relative to the common and unique variance, that is, the proportion of the non-error variance), is a relatively high 0.9 (0.886). This indicates that the subscales were, in general, correlated sufficiently highly that, together with the four subscales, the greatest component of variance is the true, common variance. …….. This high proportion of true common variance of the scale suggests that for the purposes to which ASAT was put, using a single score was generally justifiable.

Thus they need to review their findings and the implications for the analysis that follows.

Response: Thank you for committing on this interpretation and we fully agree upon the reviewer’s comment. In the new major revised Result section the interpretation is revised following the new analysis of the five-dimension without item-reduction and the indices are analysed together with the result of the PCA/t-test protocol (Page 21-22, line 342-9).

Reviewer Comment 11:

Line 200: There is some risk here that any observed multidimensionality is simply the result of local dependency. Indeed, at this stage it is becoming unclear as to which item sets are grouped. Does this analysis follow from the 17 item set, or dimension-based testlets.

Response: Thank you for committing on this and requesting clarification. In the substantially revised Method and Result sections, we hope the new structure clarifies the logic of how item sets were handled. Table 6 ( page 17) has been added to describe in detail which items belonged to each subscale, and the “Step two” section explains how we progressed from the individual subscale analyses to the subtest analysis, where items within each subscale were combined into a subtest (Page 20, line 326-329).

Reviewer Comment 12:

210. As the items have not yet been introduced they need explicit identification.

Response: Thank you for pointing this out. Table 2 on page 12 has been added to clearly state all items for explicit identification.

Reviewer Comment 13:

Line 222: It would be helpful if it was made clear that the analysis following related to the item level analysis of the 17 item scale.

Response: Thank you for pointing this out and we hope that the major revised structure of the Method and Result sections are helpful in clarifying the different steps of the analysis and which item analysis that belonged to which scale.

Reviewer Comment 14:

Line 225: A Table reporting chi-square and residual fit statistics for the 17 items should be provided to give the reader the staring point for the analysis. Also the summary item-trait interaction Chi-Square and reliability should be reported here, and also for each subsequent analysis revision.

Response: Thank you for your suggestion to improve the reporting of Chi-square and residual fit statistics. The following tables have been added to present Chi-square and residual fit results for all analyses:

Table 5, p.15 – item fit statistics original 17-item scale

Table 6, p.17 – item fit statistics 5C subscales

Table 7, p.21 - item fit statistics subtest the five-dimensions

Table 8, p. 23 - item fit statistics subscales Self-worth and Pro-social

Table 9, p. 27 - item fit statistics subtest of the two-dimensions

The summary item-trait interaction chi-square as well as the PSI (Cronbach alpha for subtests) have been added in the texts for all the different analyses in the new major revised Result section.

Reviewer Comment 15:

Line 231: These three items belong to the same subscale, indicating potential LD

Response: Many thanks for highlighting this and we have now further highlighted the LD in the revised Discussion section (Page 28, line 486-496).

Reviewer Comment 16:

Line 234: Removal of items should normally be a last resort but in light of the explanation given, this may be correct in this instance. As they belong to different subscales their removal should not compromise individual subscale analysis. However, it would have been informative if they also showed serious misfit within each subscale, which would have added to the veracity of the decision to remove them. Furthermore, individual item fit may be confounded by the presence of LD and the subscale level analysis may help inform on this.

Response: Many thanks for your wise input and the updated Method and Result sections now includes all items, hence no item reduction takes place in the updated manuscript.

Reviewer Comment 17:

237. The problem with item 5 should be picked up in the discussion as it reflects on the adaptation process, i.e. double negatives should not have arisen.

Response: Thanks for your feedback. This issue is now highlighted in the Discussion section. The double negative was present in the original English scale and had not previously been adressed; therefore we added a general discussion on item phrasing and how it may influence participants’ responses (Page 30-31, line 550-557).

Reviewer Comment 18:

Line 244: Is this related to subtest analysis within each subscale, or at the subscale level for the whole scale? For the interpretation of the result, see the comment at 195 above.

Response: Many thanks for pointing out the need for clarification. The new structure of the updated Method and Results sections is intended to make it clearer how each analysis relates to the previous steps and whether the results concern subtest or subscale levels. We hope this updated presentation helps the reader follow the analytical logic throughout the manuscript.

Reviewer Comment 19:

Line 254: Give the above (195) the multidimensionality conclusion is not correct, but the residual correlations of the subtest is informative, and the subsequent two domain result

Response: Many thanks for highlighting this. As the Method and Result sections have undergone major revision, so have the Discussion section as well where we argue for our combined analysis of the indices together with the results from the PCA/t-test protocol (Page 28-29, line 498-508).

Reviewer Comment 20:

Line 262-263: The two domains, taken, together, have been shown to fit the Rasch model. Essentially the testlets have absorbed the LD and its influence upon individual domain fit. Using the test-equating procedure on RUMM2030 the authors could now produce a transformation table for the total score, and each domain, all on the same metric (reminding readers that two items from the 17 were removed).

The issue then arises, why try and analyze the item set of each domain without resolving any of the problems that arise? This leaves the readers with the conundrum of having seen the two domains emerge in the testlet analysis with apparently good fit etc., but now showing poor fit at the item level! For purposes of revealing the reliability of the new domain, this may be best resolved in future analysis involving a test-retest situation

Response: Many thanks for your detailed and insightful comments regarding the overall analysis and our conclusion about the final two-dimensional solution. In the revised Methods section, no item reduction was performed, and additional analyses were conducted to increase transparency and to further explore the LD and DIF within the subscales. Based on these results, we continue to support the two-dimensional solution, and transformation tables of have been produced for each domain. As the hierarchical order observed in the analysis (Table 9, page 27) suggested that Self-worth appears to be a prerequisite for Pro-social, it seems more meaningful to treat the two dimensions separately rather than combining them into a single total score. This distinction is also more meaningful from both a research and a clinical/practical perspective, as it allows for more nuanced interpretation and application of the findings (Page 29, line 527-531).

Reviewer Comment 21:

343. Where is the results for item fit of each subscale which supports this conclusion?

Response: Many thanks for highlighting the lack of referring to relevant item fit statistics. As the Method and Results sections are updated and new item-fit statistics is presented in Tables this particular result is no longer part of the manuscript.

Reviewer Comment 22:

378. This implies satisfactory fit of each domain, yet the analytical conclusion is one of misfit

Response: Many thanks for highlighting this contradiction. As the Method and Results sections are updated new interpretations are presented in the new manuscript.

Reviewer Comment 23:

More information should be provided about the translation process. What were the competences of these 6 lay people? How was the translation process carried out? How many reviewers translated the version from English to Swedish and how many people from Swedish to English? What was the composition of the expert panel that evaluated the translations? How was the final Swedish version developed? Report the demographic characteristics of the 15 pupils involved in this process.

Response: Many thanks for your feedback on clarifying the translation process. Additional demographic data have been provided for both the panel and the pupils. The six lay people were from outside academia, and their involvement aimed to reduce the level of technical language, as recommended by Hunt. As described by Hunt, cultural adaptation is an iterative process in which forward and backward translations within the bilingual panel were conducted continuously until the most acceptable translation was agreed upon. Since the goal was not a not a word-for-word translation but rather to capture the same phenomena as origin

---

## [Decision Letter · Decision Letter 1]

16 Dec 2025

Exploring the dimensionality of the 5C Positive Youth Development very short form using Rasch Measurement theory in Swedish upper secondary school contexts.

PONE-D-25-20352R1

Dear Dr. Bergman,

We’re pleased to inform you that your manuscript has been judged scientifically suitable for publication and will be formally accepted for publication once it meets all outstanding technical requirements.

Kind regards,

Karl Bang Christensen, Ph.D.

Academic Editor

PLOS One

Additional Editor Comments (optional):

Reviewers' comments:

Reviewer's Responses to Questions

**Comments to the Author**

Reviewer #1: All comments have been addressed

2. Is the manuscript technically sound, and do the data support the conclusions?

Reviewer #1: Yes

3. Has the statistical analysis been performed appropriately and rigorously?

Reviewer #1: Yes

4. Have the authors made all data underlying the findings in their manuscript fully available?

Reviewer #1: Yes

5. Is the manuscript presented in an intelligible fashion and written in standard English?

Reviewer #1: Yes

Reviewer #1: 464 Table 9 is, in effect, the key findings of the study. A testlet based analysis of the two conceptual groupings. Their justification for retaining the two-dimensional solution, albeit when the reported result is essentially a unidimensional solution retaining 80% of the variance, is nevertheless valid, as it will clearly inform further exploration of the relationship between the two domains and potentially inform interventions to improve PYD.

613 Conclusion – very well written and informative.

**Do you want your identity to be public for this peer review?** For information about this choice, including consent withdrawal, please see our Privacy Policy

Reviewer #1: No

---

## [Editor Report · Acceptance letter]

PONE-D-25-20352R1

PLOS One

Dear Dr. Bergman,

I'm pleased to inform you that your manuscript has been deemed suitable for publication in PLOS One. Congratulations! Your manuscript is now being handed over to our production team.

Kind regards,

on behalf of

Dr. Karl Bang Christensen

Academic Editor

PLOS One